# The human coronavirus HCoV-229E S-protein structure and receptor binding

Zhijie Li[1], Aidan CA Tomlinson[2], Alan HM Wong[2], Dongxia Zhou[1], Marc Desforges[3], Pierre J Talbot[3], Samir Benlekbir[4], John L Rubinstein[2,4,5], James M Rini[1,2]*

[1]Department of Molecular Genetics, The University of Toronto, Toronto, Canada; [2]Department of Biochemistry, The University of Toronto, Toronto, Canada; [3]Laboratory of Neuroimmunovirology, INRS-Institut Armand-Frappier, Institut National de la Recherche Scientifique, Université du Québec, Laval, Canada; [4]Molecular Medicine Program, The Hospital for Sick Children Research Institute, Toronto, Canada; [5]Department of Medical Biophysics, The University of Toronto, Toronto, Canada

**Abstract** The coronavirus S-protein mediates receptor binding and fusion of the viral and host cell membranes. In HCoV-229E, its receptor binding domain (RBD) shows extensive sequence variation but how S-protein function is maintained is not understood. Reported are the X-ray crystal structures of Class III-V RBDs in complex with human aminopeptidase N (hAPN), as well as the electron cryomicroscopy structure of the 229E S-protein. The structures show that common core interactions define the specificity for hAPN and that the peripheral RBD sequence variation is accommodated by loop plasticity. The results provide insight into immune evasion and the cross-species transmission of 229E and related coronaviruses. We also find that the 229E S-protein can expose a portion of its helical core to solvent. This is undoubtedly facilitated by hydrophilic subunit interfaces that we show are conserved among coronaviruses. These interfaces likely play a role in the S-protein conformational changes associated with membrane fusion.

*For correspondence:
james.rini@utoronto.ca

Competing interests: The authors declare that no competing interests exist.

## Introduction

Coronaviruses are enveloped RNA viruses found in mammals and birds (*Graham et al., 2013*; *Su et al., 2016*). They share a common ancestor thought to have originated in bats (*Woo et al., 2012*) and several genera that use a wide range of protein and carbohydrate receptors have evolved (*Forni et al., 2017*; *Peck et al., 2015*). Four coronaviruses, HCoV-229E, HCoV-NL63, HCoV-OC43 and HCoV-HKU1, circulate in the human population where they are responsible for approximately one-third of the common cold (*Gaunt et al., 2010*; *Lim et al., 2016*). HCoV-229E and HCoV-NL63 are closely related alphacoronaviruses that use different host proteins as receptors (*Yeager et al., 1992*; *Hofmann et al., 2005*), while HCoV-OC43 and HCoV-HKU1 are betacoronaviruses that both use 9-O-acetylsialic acids as receptor (*Vlasak et al., 1988*; *Hulswit et al., 2019*). HCoV-229E-like coronaviruses have recently been identified in bats (*Corman et al., 2015*), camels (*Sabir et al., 2016*; *Corman et al., 2016*) and alpacas (*Crossley et al., 2012*), an indication that transmission from bats to humans may have involved an intermediate host.

HCoV-229E uses human aminopeptidase N (hAPN) as receptor (*Yeager et al., 1992*) and it was recently shown in a cell-based entry assay that the camel 229E-like CoV can also use hAPN (*Corman et al., 2016*). Although these viruses might bind their host APNs in a structurally conserved fashion, it should be noted that the porcine alphacoronavirus, PRCoV, binds to a site on porcine APN that differs from the site at which HCoV-229E binds to hAPN (*Reguera et al., 2012*;

*Wong et al., 2017*). Moreover, the more closely related human alphacoronavirus, HCoV-NL63, uses angiotensin converting enzyme 2 as receptor (*Hofmann et al., 2005*). These observations suggest the existence of mechanisms for acquiring new receptor interactions, whether conserved or non-conserved, an important step in cross-species transmission. The possibility that camels were involved in the transmission of HCoV-229E to humans is interesting given that the highly virulent MERS-CoV is directly transmitted to humans from dromedaries in many cases (*Paden et al., 2018*).

The coronavirus S-protein is responsible for both host receptor binding and fusion of the viral and host cell membranes (*Li, 2016*). Its ectodomain is composed of the N-terminal S1 region that harbors the receptor binding domain (RBD) and the C-terminal S2 region that mediates membrane fusion. The S-protein is a trimer in both the pre-fusion and post-fusion conformations and cryo-EM analysis has provided structures for both conformations (*Kirchdoerfer et al., 2016*; *Walls et al., 2016a*; *Kirchdoerfer et al., 2018*; *Walls et al., 2016b*; *Walls et al., 2017*; *Gui et al., 2017*; *Yuan et al., 2017*; *Song et al., 2018*; *Shang et al., 2018a*; *Shang et al., 2018b*; *Tortorici et al., 2019*). Notably, the RBD has been found to access two different conformations in the pre-fusion form of some coronavirus S-proteins (*Kirchdoerfer et al., 2018*; *Gui et al., 2017*; *Yuan et al., 2017*; *Song et al., 2018*). In the down conformation, the receptor binding site is blocked, and it has been suggested that this may serve to protect it from immune recognition (*Walls et al., 2016a*; *Gui et al., 2017*). In the up conformation, it is exposed for receptor binding and data exist to suggest that the RBD up conformation may promote conversion to the post-fusion form (*Walls et al., 2019*). The post-fusion form of the coronavirus S-protein is characterized by a 6-helix bundle formed by the inner HR1 triple-helical coiled-coil around which the HR2 helices are packed (*Xu et al., 2004*; *Yan et al., 2018*; *Zhang et al., 2018*). The coronavirus S2 subunit is large and the structural changes that accompany conversion to the post-fusion conformation are extensive (*Walls et al., 2017*). Indeed, the HR1 triple-helical coiled-coil is formed entirely on conversion to the post-fusion form, and the HR2 helices, which form a triple-helical coiled-coil in the pre-fusion conformation, must dissociate before participating in 6-helix bundle formation. The S-protein structural features that enable these dramatic conformational changes are not yet fully described.

We previously reported the structure of the receptor binding domain of HCoV-229E in complex with hAPN (*Wong et al., 2017*). The structure showed that three extended RBD loops are solely responsible for receptor binding. Moreover, we showed that these loops are the most variable region in the entire viral genome. Indeed, a phylogenetic analysis defined six RBD classes whose viruses have successively replaced each other in the human population over the past 50 years. The RBD classes differ in their ability to be bound by a receptor binding loop-specific neutralizing antibody, and in their affinity for hAPN, two determinants of viral fitness. Although immune evasion provides an explanation for the emergence of these RBD classes, how extensive loop variation is accommodated while maintaining receptor binding remained an unanswered question. Loop variation is also expected to facilitate the acquisition of new receptor interactions and thereby cross-species transmission, and how this might work in structural terms is also not well understood.

To address these questions and to gain further insight into how the 229E S-protein mediates membrane fusion, we determined the X-ray crystal structures of three additional RBD (Classes III, IV and V) complexes with hAPN, as well as the cryo-EM structure of the ectodomain of the S-protein trimer. We find that an invariant set of core interactions defines the interaction with hAPN across all the RBD classes characterized. Receptor binding loop variation is concentrated in regions peripheral to the core and it is accommodated by the inherent plasticity of extended loops. The results have provided a model for how HCoV-229E could evade receptor loop-binding neutralizing antibodies while maintaining receptor binding. They also support the suggestion that the 229E-like bat, camel and alpaca coronaviruses use APN as receptor and bind it essentially as shown for our RBD-hAPN complexes. The structures also show that receptor binding loop variation is unlikely to modulate the RBD up/down equilibrium or the stability of the pre-fusion trimer. As shown by the cryo-EM structure, the 229E S-protein can access a conformation that exposes a significant portion of its helical core region to solvent. The ability to do so likely stems from the fact that the interfaces between monomers in this region are hydrophilic, a property which we now show is shared by all the coronavirus S-proteins whose structures have been determined. These hydrophilic interfaces likely play a role in the conversion of the coronavirus S-protein from its pre-fusion to its post-fusion conformation during the process of membrane fusion.

## Results

### 229E receptor binding loop variation and receptor recognition

Our previously reported RBD-hAPN complex used the RBD of a lab-adapted strain of 229E (ATCC strain VR-740, Class I) (*Wong et al., 2017*). To gain insight into how loop variation is accommodated while maintaining receptor binding, we have now determined the structures of the Class III-V RBDs in complex with hAPN. These RBDs possess the receptor binding loops (Loop 1, residues 308–325; Loop 2, residues 352–359; Loop 3, residues 404–408) and supporting residues of their respective RBD classes and the Class I RBD residues in the remainder of the domain as previously described (*Figure 1*, *Figure 1—figure supplement 1*) (*Wong et al., 2017*).

With reference to the Class I complex, the structures show that the interactions between Loop 1 residues 315–320 and hAPN residues 287–291 are highly conserved among all four complexes (*Figure 1b,c*). There are six hydrogen bonds between Loop 1 and hAPN that are common to all the complexes. Of these, four involve backbone atoms of Loop 1 and five involve backbone atoms of hAPN. Indeed, the 287–291 segment of hAPN is a surface-exposed β-strand and these hydrogen bonds satisfy all its exposed backbone amide and carbonyl groups. These common hydrogen bonds also involve the side chains of Loop 1 residue Asn 319 and hAPN residue Asp 288. The importance of these core interactions is underscored by the fact that Gly 315, Cys 317, Asn 319, and Cys 320 correspond to four of the six residues that are absolutely conserved among the receptor binding loops of all 229E isolates sequenced (*Wong et al., 2017*) (*Figure 1d*, *Figure 1—figure supplement 1*). Gly 315 is in the $\alpha_L$ region of φ/ψ-space, a backbone conformation only easily accessed by glycine. Cys 317 and Cys 320 form a disulfide bond which constrains the loop conformation required to satisfy the observed hydrogen bonds. The side chain of Asn 319 is buried in the complex interface where it donates and accepts hydrogen bonds from the backbone atoms of hAPN residue Glu 291. The stacking interaction between the backbone of residue Cys 317 and the side chain of hAPN residue Tyr 289 also plays an important role. Comparison of these four structures establishes that these core interactions underpin 229E's specificity for hAPN. Gly 313 is also absolutely conserved among all 229E isolates sequenced and its backbone conformation varies among the complexes. It seems that the flexibility afforded by glycine is important in maintaining the core Loop 1 interactions with hAPN in the context of the sequence variation that occurs in the three receptor binding loops outside of the core region.

RBD Classes III, IV, and V are from viruses that arose while circulating in the human population and they share important features that differentiate them from the tissue culture-adapted Class I virus (*Figure 1a*). In Loop 2, they all have a deletion of two amino acids that allows Arg 357 of Loop 2, the last of the absolutely conserved residues, to form a stronger salt bridge with Asp 315 of hAPN than what was observed in the Class I complex. In Loop 3, all three classes have acquired two leucine residues not found in the Class I sequence: Trp 404 and Ser 407 of Class I are replaced by Leu 402 and Leu 405 in Classes III, IV and V. The sidechains of these leucine residues occupy the volume filled by the tryptophan sidechain in the Class I complex. As a result, Leu 405 makes the apolar contacts with Leu 318 of hAPN, a role played by Trp 404 in the Class I complex. The sequence changes in Loops 2 and 3 lead to a small re-orientation of the main body of the RBD in the Class III, IV and V complexes. This in turn leads to a compensatory twist in Loop 1 which preserves the core Loop 1-hAPN interaction. As a result of the twist, Arg 316 emerges from a different side of Loop 1 where it forms a salt bridge with Asp 288 of hAPN in the Class III, IV and V complexes, which is not possible in the Class I complex.

Relative to Class I, there are also notable differences specific to only one or two of the other classes. The GGG motif (313-315) at the tip of Loop 1 in Class I is a GVG in Classes III and IV; the presence of the valine residue (Val 314) leads to an increase in the van der Waals contact with hAPN. In Class V, the motif is GPG and by constraining the Loop 1 conformation, the proline residue (Pro 314) likely leads to a stabilization of the interactions with hAPN. In Classes IV and V, the conversion of Gln 311 to Arg 311 permits a weak salt bridge interaction with Glu 291 that is not possible in Classes I or III. Overall, the structures show that sequence variation is concentrated in the peripheral regions where it is accommodated by loop plasticity. Importantly, this plasticity also serves to ensure that the core interactions which define the specificity for hAPN are preserved.

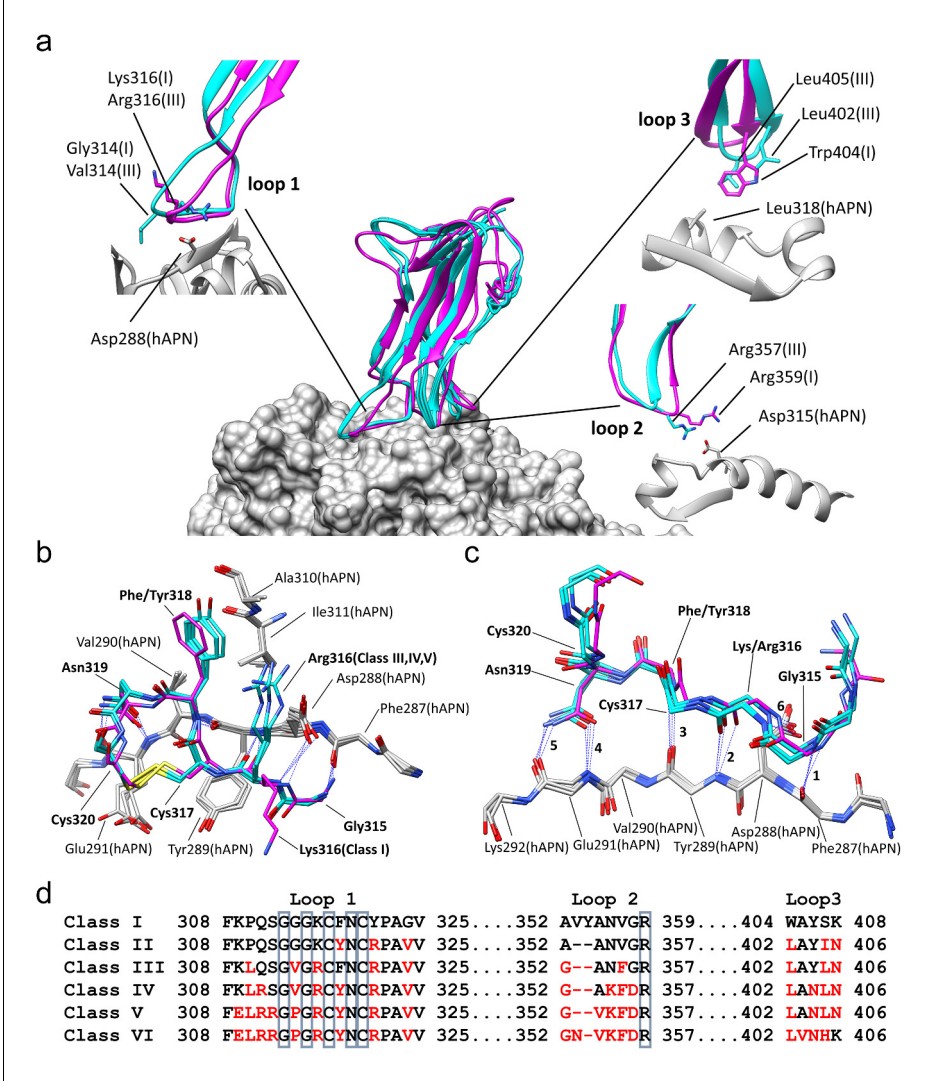

**Figure 1.** Comparison of the RBD-hAPN complexes. In all panels, the Class I RBD is colored magenta, the Class III, IV and V RBDs are colored cyan, and hAPN is colored gray. (a) Center: superimposition of the Class I, III IV and V RBD-hAPN complexes. Surrounding: the Class I and Class III RBDs are shown; the Class III RBD provides an example of the major differences observed between the Class III, IV and V RBDs relative to Class I. Loop 1, a twist in the loop allows Arg316 to interact with Asp288. Loop 2, a 2-aa deletion reorients Arg357, enabling a better interaction with Asp315. Loop 3, Leu402 and Leu405 replace Trp404, with Leu405 maintaining the apolar contact with Leu318. (b) The 315–320 segment of Loop 1shows a high degree of structural conservation among the four complexes. (c) Six hydrogen bonds (blue dotted lines, marked 1–6) are conserved among the four complexes and they all involve Loop 1 residues 315–320. Five of them (hydrogen bonds 1–5 in the figure) involve the five exposed backbone amide/carbonyl groups of hAPN residues 287–291; the sixth is formed with the sidechain of hAPN residue Asp 288. For clarity, only sidechains that participate in hydrogen bonds are shown. (d) Receptor binding loop sequence alignment. Differences relative to Class I are colored red. The six residues conserved among all HCoV-229E viruses sequenced are enclosed in blue boxes.

The online version of this article includes the following figure supplement(s) for figure 1:

**Figure supplement 1.** RBD sequence Alignment.

## Negative stain and Cryo-EM analysis

To provide further insight into how the 229E S-protein mediates receptor binding and ultimately membrane fusion, we determined the structure of its ectodomain by single-particle cryo-EM analysis. To facilitate expression and structure determination, the ectodomain (residues 17–1113) was

produced as a fusion protein with a C-terminal fibritin foldon trimerization motif (*Tao et al., 1997*). In addition, a dual-proline mutation (Thr871Pro/Ile872Pro) was introduced to stabilize the pre-fusion form as previously described (*Pallesen et al., 2017*). Negative stain EM analysis showed that the 229E S-protein consistently appeared as uniformly sized, cone-shaped particles in the micrographs (*Figure 2a*). A flexible tail that is presumed to be a combination of the HR2 region and the trimerization motif is also visible in the micrographs. The 2D class averages from these micrographs further showed that the cones have domain-sized bulges and an opening at their bottom end (*Figure 2c*).

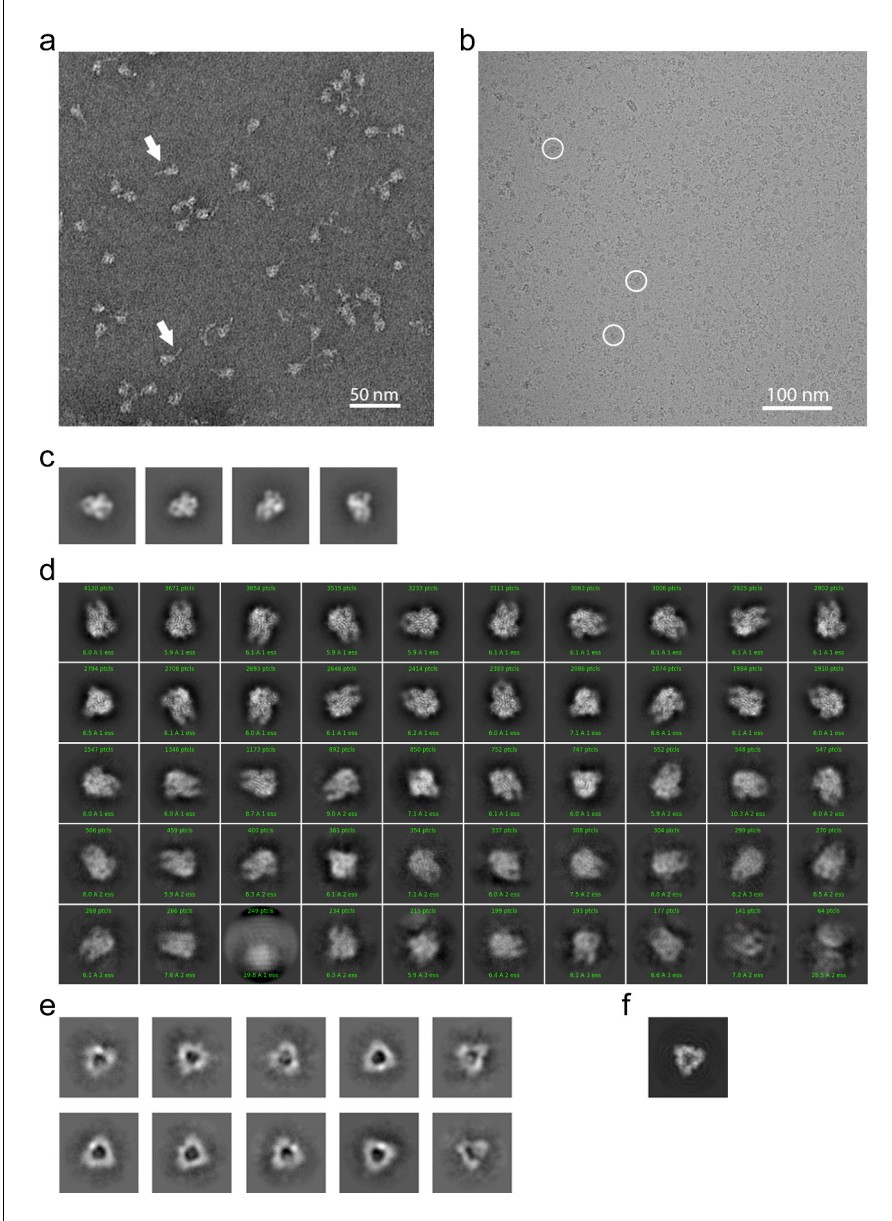

**Figure 2.** EM analysis of the HCoV-229E S-protein. (**a**) An example negative stain micrograph. White arrowheads indicate particles with a readily discernable tail which is presumed to be formed by the HR2 region and the added foldon trimerization domain. (**b**) An example cryo-EM micrograph. For reference, three 229E S-protein particles are enclosed by white circles. (**c**) Selected 2D class averages of negative stain particles. (**d**) Cryo-EM 2D class averages of the ~71,000 particles used in the final refinement. (**e**) Cryo-EM 2D class averages of the putative S1 cap. (**f**) A simulated top view of the S1 cap at the same scale as in (**e**).

The online version of this article includes the following figure supplement(s) for figure 2:

**Figure supplement 1.** Orientation distribution of the 229E S-protein cryo-EM data.

These features are also observed in the cryo-EM structure and they result from an opening of the monomers in the helical core and connector region of the trimer, as discussed below.

For cryo-EM structure determination, freshly prepared 229E ectodomain was frozen on EM grids and analyzed with single-particle techniques (*Figure 2b*). During data processing, extensive 2D and 3D classification were performed to remove a large number of small particles that did not represent the intact complex. The final dataset included ~71,000 particle images, most of them showing side-views of the trimeric S-protein (*Figure 2d*, *Figure 2—figure supplement 1*). Since refinement with C1 symmetry did not reveal particle classes lacking three-fold symmetry, C3 symmetry was imposed in the final cryo-EM map and during 3D refinement. This led to a final map with a resolution of 3.1 Å (*Figure 3*, *Figure 3—figure supplement 1*, *Supplementary file 1*).

The cryo-EM images also produced 2D class averages corresponding to particles shaped like equilateral triangles with ~110 Å edges (*Figure 2e*). These class averages have little density in the center of the triangle and appear to correspond to the S1 region of the trimer alone (*Figure 2f*). Since SDS-PAGE analysis shows no evidence for proteolysis of the sample, it seems that the S2 region is present but disordered in these particles. These 2D class averages did not show internal structural detail and attempts to generate high-resolution 3D reconstructions were not successful. Nevertheless, their presence suggests that the S1 subunits can maintain a trimeric arrangement even when the S2 subunits are disordered.

## Overall structure of the 229E S-protein trimer

The overall structure of the 229E S-protein trimer is similar to that of the closely related NL63 S-protein trimer (*Walls et al., 2016a*). The three S1 subunits form a cap over the three S2 subunits with each S1 subunit non-covalently associating with the S2 subunit of an adjacent monomer (*Figure 4*). The S1 subunit of each monomer is composed of four β-rich domains: A, B, C and D (*Figure 4c,d,e*). The S2 subunit is largely alpha-helical and the three longest helices, helix 629–652 (UH), helix 821–

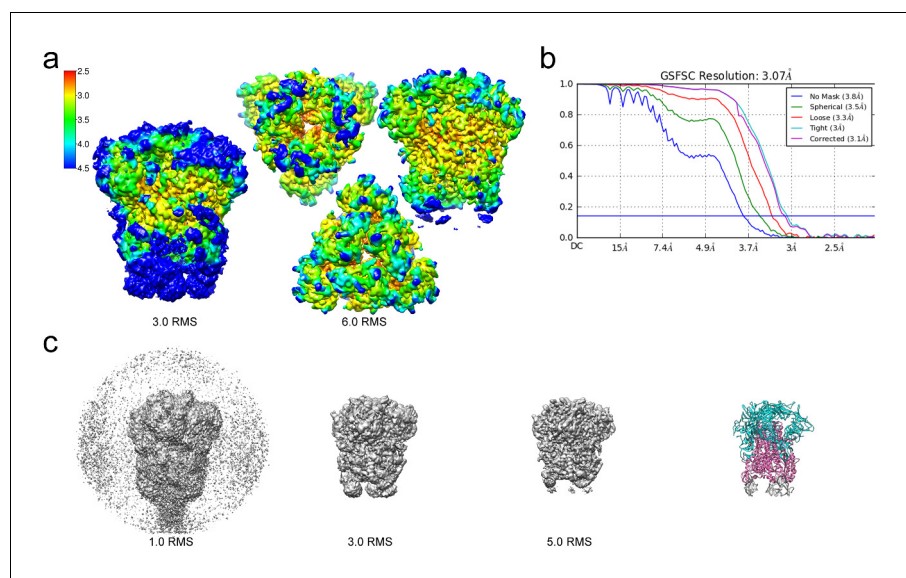

**Figure 3.** Resolution of the HCoV-229E S-protein cryo-EM map. (**a**) Local resolution (Å) plotted on the cryo-EM map surface as a heat map. Leftmost image, a side view of the map contoured at 3.0 RMS to show the weak densities of the connector domains (at bottom). To the right, three views of the map contoured at 6.0 RMS showing that the entire S1 region and the helical core region of S2 are well-resolved. (**b**) GSFSC curves of the final 3D non-uniform refinement from cryoSPARC v2. The blue horizontal line indicates an FSC value of 0.143. (**c**) The cryo-EM map contoured at different levels to show the weak densities near the C-terminal end (bottom). To the right, the atomic model is shown in ribbon representation with the well resolved regions of the S1 and S2 subunits colored cyan and magenta, respectively. The poorly resolved connector domains are colored gray.

The online version of this article includes the following figure supplement(s) for figure 3:

**Figure supplement 1.** Examples of the atomic model built into the 229E S-protein cryo-EM map.

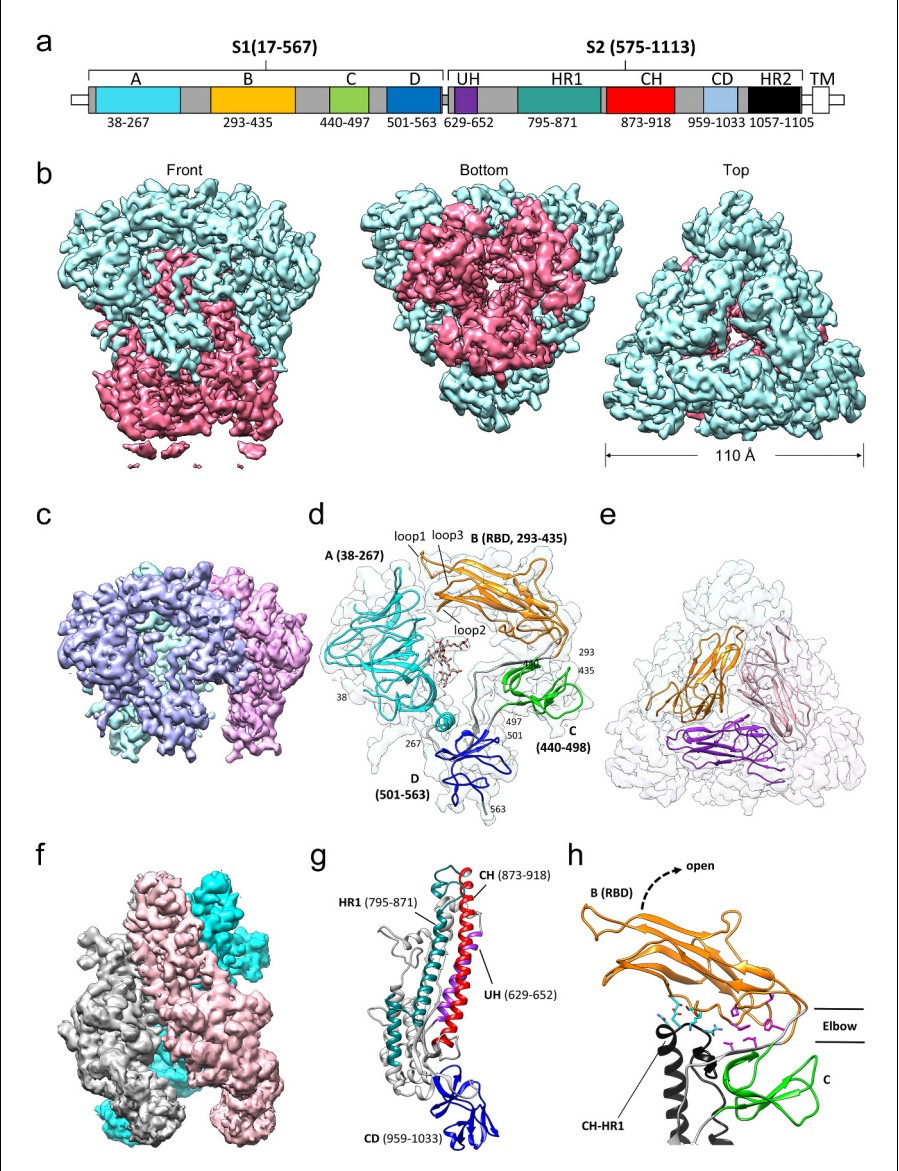

**Figure 4.** Cryo-EM structure of the HCoV-229E S-protein. (**a**) Domain structure of the S-protein. UH, upstream helix; HR1, heptad repeat 1; CH, central helix; CD, connector domain; HR2, heptad repeat 2. TM, transmembrane segment. Segments colored in white are not part of the expressed ectodomain construct. (**b**) Three views of the unsharpened cryo-EM map contoured at 6.0 RMS. The S1 and S2 regions are colored cyan and magenta, respectively. (**c**) The portion of the cryo-EM map corresponding to the S1 region contoured at 5.0 RMS. Each subunit is colored differently. (**d**) Domain structure of an S1 subunit shown within a transparent rendering of the cryo-EM map. The highly ordered N-glycan (Asn 62) found in the center of the S1 subunit is shown in stick representation. The three receptor binding loops of the RBD (domain B) are labeled. (**e**) A portion of the cryo-EM map showing the S1 region viewed from the top. The three RBDs are shown as ribbons. (**f**) The portion of the cryo-EM map corresponding to the S2 region contoured at 3.0 RMS. Each subunit is colored differently. (**g**) Ribbon representation of the S2 subunit. (**h**) In the down conformation, the RBD (domain B, orange) makes polar interactions (cyan sidechains) with the CH-HR1 junction and hydrophobic interactions (magenta sidechains) in the elbow between domain B and domain C. The dotted arrow shows the direction that the RBD will take on conversion to the up conformation.

The online version of this article includes the following figure supplement(s) for figure 4:

**Figure supplement 1.** Fitting of the connector domain into its cryo-EM map density.

852 (HR1$_{821-852}$) and helix 873–918 (the central helix, CH), form the helical core region of each S2 subunit (*Figure 4f,g*). The HR1$_{821-852}$ helix is only a portion of the HR1 segment that forms the central 3-helix coiled-coil of the post-fusion 6-helix bundle (*Yan et al., 2018*; *Zhang et al., 2018*). The remainder of HR1 is made up of two shorter helices and two non-helical segments (*Figure 4g*). The junction between the HR1 helices and the start of CH is a sharp turn that reverses the chain direction leading to the anti-parallel arrangement of the HR1$_{821-852}$ and CH helices. The importance of the HR1-CH junction stems from the fact that in the post-fusion form it is restructured to form a connection between the HR1 and CH helices which now form a continuous helix (*Figure 5b,c*). It is at this site that the Thr871Pro/Ile872Pro mutations have been introduced to stabilize the pre-fusion conformation. The ~80 residue (residues 959–1033) β-rich connector domain (CD) could be identified in the cryo-EM map, but its density was too weak to be traced with confidence (*Figure 3c*). In this region, a homology model based on the CD domain of the NL63 S-protein was positioned in the density (*Figure 4f,g*, *Figure 4—figure supplement 1*). This extended our model to 24 residues before the start of the HR2 region.

The 229E S-protein contains as many as 30 predicted N-glycosylation sites on each monomer (*Figure 5a*). Eighteen of these are located in the well-resolved S1 and S2 regions (38–565 and 585–953) and clear N-glycan density is observed for 17 of them. Although not observed in our cryo-EM structure, the ~60 residue HR2 region possesses 5 N-glycosylation sequons. As such, it represents the most highly glycosylated region of the entire S-protein, a feature that may be important in the transition from the pre-fusion to the post-fusion conformation. The HR2 region forms a triple helical coiled-coil in the pre-fusion form (*Hakansson-McReynolds et al., 2006*) and its helices must dissociate to participate in 6-helix bundle formation. Exposure of the core apolar residues of the HR2 coiled-coil to solvent would lead to a decrease in solubility and the potential for non-specific interactions and this might be offset by the bulky and hydrophilic N-glycans. The high degree of glycosylation shown by the coronaviruses is thought to be important in shielding them from immune recognition (*Walls et al., 2016a*). Protecting the pre-fusion HR2 triple helical coiled-coil from

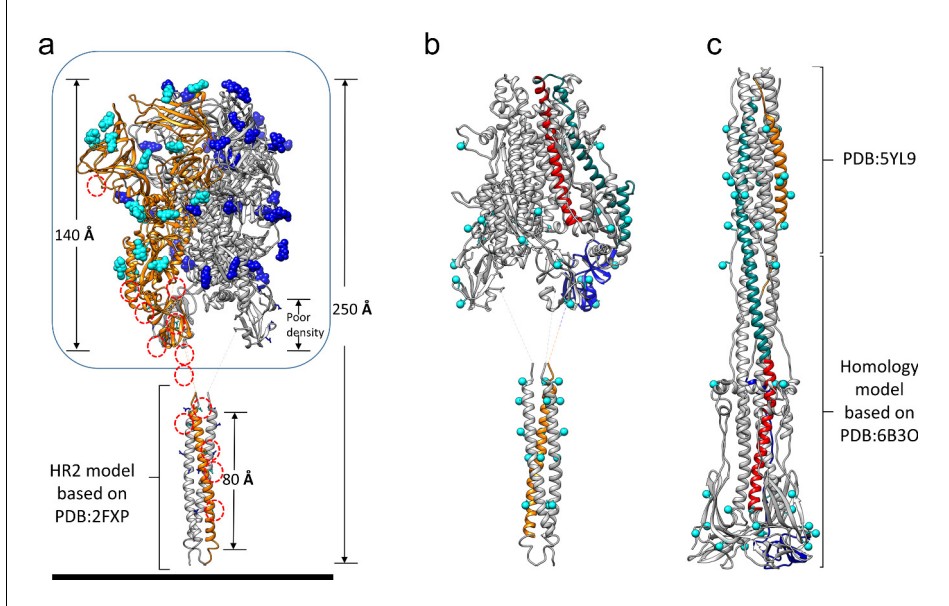

**Figure 5.** Models of the complete HCoV-229E S-protein ectodomain and locations of the N-glycosylation sites. (a) A composite model of the complete ectodomain showing the cryo-EM structure determined (enclosed by the box) and a homology model of the HR2 region based on PDB: 2FXP. For each N-glycan observed in the cryo-EM map, the Asn-linked GlcNAc moiety is shown in sphere representation; cyan, those on chain A (orange ribbons). The red dotted circles indicate other potential N-glycosylation sites on chain A. (b) The S2 region of the pre-fusion model. (c) A composite model of the 229E post-fusion form based on the MHV post-fusion structure (PDB: 6B3O) and a crystal structure of the post-fusion 229E HR1-HR2 six-helical bundle (PDB: 5YL9). Red, CH; teal, HR1; orange, HR2. Each potential N-glycan is indicated by a cyan sphere at the Asn sidechain amide group.

immune recognition is another possible explanation for the high degree of glycosylation observed in this region.

## The S1 subunit and receptor binding loop accommodation

Viewed in isolation, the four β-domains (domains A-D) of the S1 subunit are arranged in a ring-like fashion with the N-glycan at Asn 62 of domain A directed into the otherwise empty center of the ring. This N-glycan is the most well-ordered in the entire structure and density for it beyond the tri-mannose core is observed. The close approach of the three receptor binding loops of domain B (the RBD) to domain A closes the ring (*Figure 4d*). As discussed above, Loop 1 forms a conserved set of core interactions with hAPN and in all the RBD-hAPN complexes it is well ordered. However, in the S-protein trimer (which contains a Class I RBD), Loop 1 is flexible, there is little density for the tip of it, and it does not make specific contacts with domain A. Loops 2 and 3 are shorter and although observed in the cryo-EM structure they too do not make specific interactions with domain A. An overlay of the Class III, Class IV and Class V RBDs, from the hAPN complexes reported here, suggests that their receptor binding loops would also not strongly interact with domain A. These observations suggest that the variable receptor-binding loops of 229E are simply accommodated in the gap between them and domain A. That the domain A surface near the loops is relatively well conserved among 229E isolates supports this suggestion as compensatory sequence changes in domain A might be expected if specific interactions with residues of the constantly evolving receptor binding loops were important.

## The RBD must flip up to bind receptor

The receptor binding domains of a number of coronaviruses have been found to access both an up and a down conformation (*Kirchdoerfer et al., 2018*; *Gui et al., 2017*; *Yuan et al., 2017*; *Song et al., 2018*). Receptor binding is only possible when domain B (the 229E RBD) is in the up conformation as the receptor binding end of domain B is blocked in the down conformation. In our cryo-EM structure, domain B is in the down conformation in all three monomers of the trimer and we see no evidence of molecules with domain B in the up conformation. Nevertheless, the structure provides a model for how the up/down conformational conversion is mediated in 229E. As shown in *Figure 4h*, the N- and C-termini of domain B are close in space and the polypeptide segments that lead into (V295-Y296-H297) and out of (V435-S436-S437) it, form a two-stranded hinge between domain B and domain C. Rotating domain B into the up conformation leads to an opening of the S1 ring and exposure of the receptor binding loops to solvent. In the down conformation, there are significant interactions between domains B and domain C of the same monomer and between domain B and the top of the S2 subunit of an adjacent monomer. The latter involves contacts with the HR1-CH junction, the connection between the HR1 and CH helices that is dramatically restructured on conversion to the post-fusion 6-helix bundle. Conversion between the up/down conformation would result in the loss/gain of both the domain B-domain C and the domain B-S2 interactions. Judging by the weak interactions between the receptor binding loops of domain B and domain A, as discussed above, the domain B-domain C and domain B-S2 interactions are the main interactions that would stabilize domain B in the down conformation.

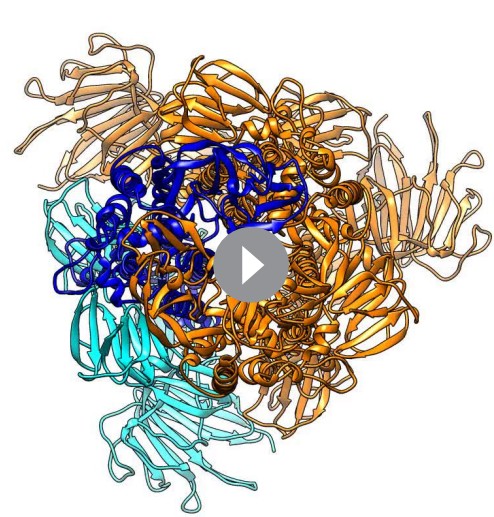

**Video 1.** Comparison of the 229E and NL63 S-proteins (Bottom view). Two views showing a morphing between the two S-proteins. PDB ID: NL63, 5SZS. An S1 subunit is colored cyan. The S2 subunit with which it associates is from another monomer and it is colored blue.
https://elifesciences.org/articles/51230#video1

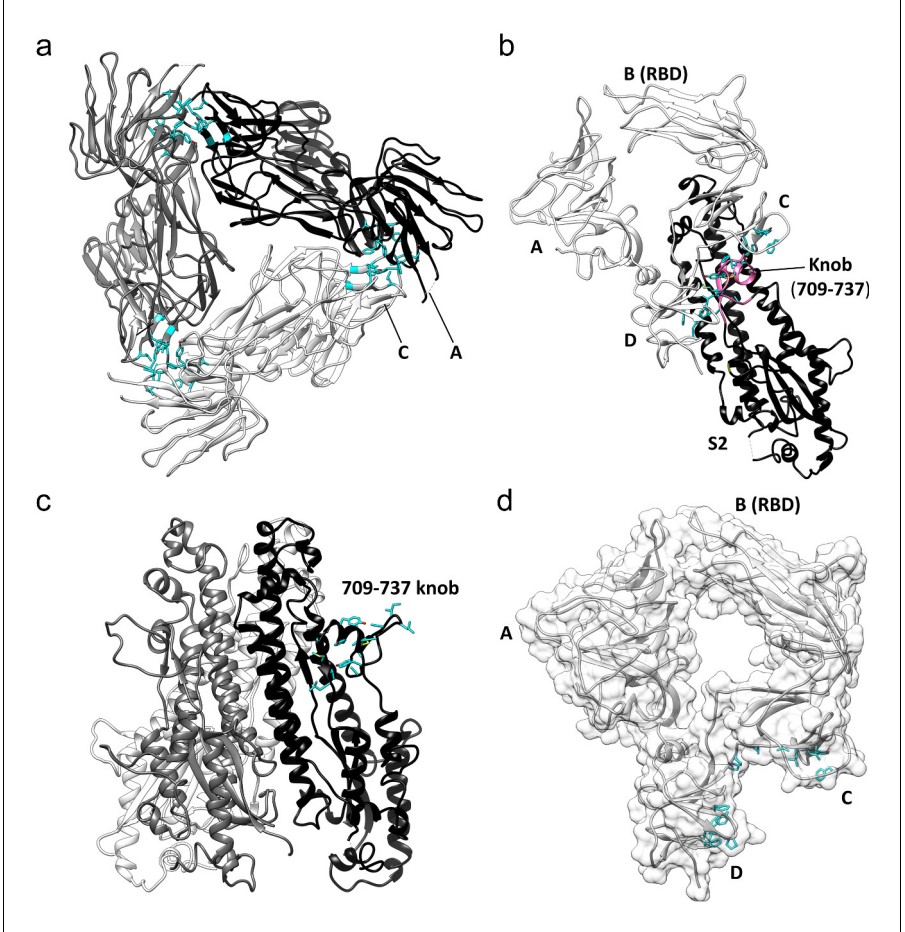

**Figure 6.** Key hydrophobic interactions defining the assembly of the HCoV-229E S-protein trimer. (**a**) Hydrophobic interactions in the S1 region. Each monomer is colored white, gray and black, respectively. Apolar residues between the S1 subunits are colored cyan. (**b**) The S1 subunit of one monomer (white) interacts with the S2 subunit of another monomer (black). The C and D domains of the S1 subunit form a hydrophobic clamp over the knob formed by S2 subunit residues 709–737 (magenta); interacting apolar side chains are shown in cyan. (**c**) Side view of the S2 region of the trimer showing the 709–737 knob; apolar residues are colored cyan (on one monomer only). (**d**) The hydrophobic clamp is formed by domains C and D; apolar residues in contact with the 709–737 knob are shown in cyan.

## The S1 cap and trimer stabilization

In the S-protein trimer, the three S1 subunits form a cap that sits over the three S2 subunits. The cap is triangular in shape with relatively little density along the three-fold rotation axis. The three A domains sit at the vertices of the triangle and the B domains are positioned closer to the three-fold axis. Apolar interactions between domain A of one S1 subunit and domain C of another largely account for the inter-subunit contacts (~700 Å$^2$) within the cap (*Figure 6a*). The presence of 2D class averages with dimensions similar to that of the cap alone suggests that these interactions can maintain cap assembly even in the absence of interactions involving the S2 subunit. Viewed down the 3-fold axis, domains C and D of the S1 subunit sit below domains A and B where they interact with the underlying S2 subunits.

The three core helices (UH, HR1$_{821-852}$, and CH) of each S2 subunit of the trimer are arranged around the 3-fold axis with CH positioned most closely to it. The interactions between the core helices of each S2 subunit make up the bulk of the trimer interface in this region. Remarkably, we find that few of the interface residues are apolar in nature. There is a small apolar patch around Val 891, but the remainder of the trimer interface in this region is composed of polar residues, many of them charged (*Figure 7a,b*, *Figure 7—figure supplement 1*). The hydrophilic nature of these interfaces

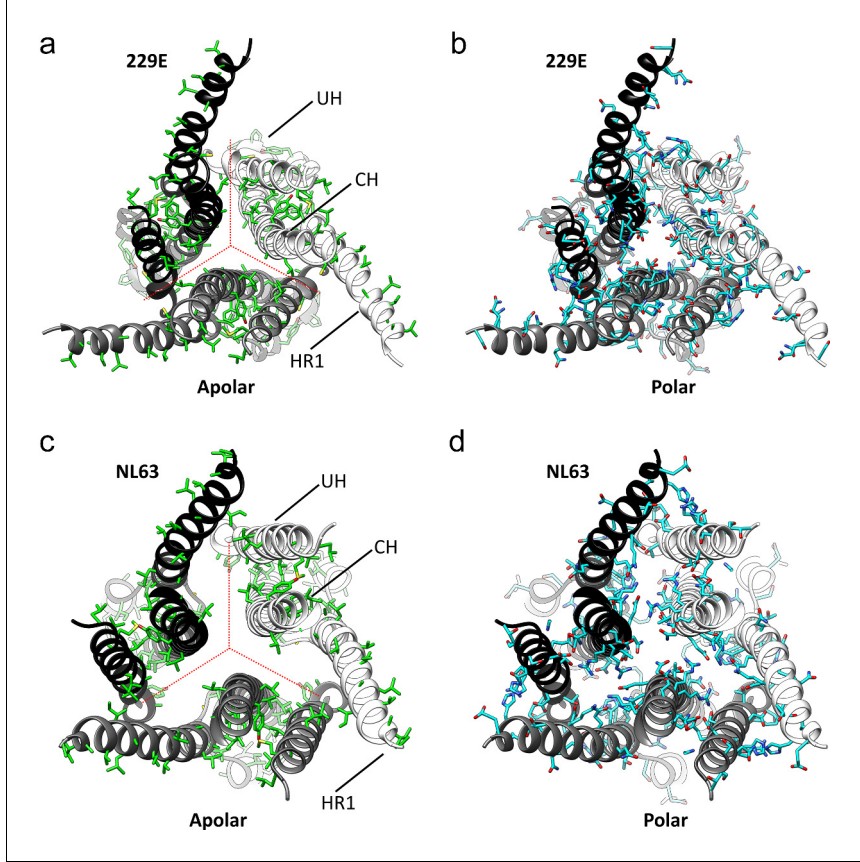

**Figure 7.** The interfaces in the S2 helical core region of the HCoV-229E S-protein are hydrophilic. (a,b) Bottom views of the interfaces in the S2 helical core region. For clarity only the three major components of the trimer interface in the S2 region are shown: UH, $HR1_{821-871}$ and CH. The red dotted lines indicate the interfaces between subunits. All the apolar residues in these segments are shown in (a). All the polar residues in these segments are shown in (b). (c, d) The trimer interfaces of the HCoV-NL63 S2 helical core region represented as in (a, b). The online version of this article includes the following figure supplement(s) for figure 7:

**Figure supplement 1.** The subunit interfaces in the helical core region are hydrophilic in coronaviruses.

suggests that the monomer-monomer contacts in the helical core region are weak. In contrast, the interactions between the S1 cap and the S2 subunits of the trimer include a relatively large apolar component capable of mediating significant stabilizing forces. Specifically, the C and D domains of each S1 subunit of the cap clamp over an apolar knob (residues 709–737) that protrudes from the helical core region of each S2 subunit. As a result, 1260 $Å^2$ of apolar surface area (of a total of ~2300 $Å^2$) is buried for each pair of S1 and S2 subunits (*Figure 6b,c,d*). As discussed above, domain B in the down conformation makes an additional contact with the S2 subunit, in this case with the critically important HR1-CH junction. Taken together, these observations suggest that the S1 cap plays a major role in the stabilization of the pre-fusion trimer. By clamping together the top end of the S2 subunits it prevents the conformational changes required for conversion to the post-fusion form. The connector domain contributes to the trimer interface in other coronavirus S-proteins and its interactions may also stabilize the pre-fusion trimer. However, as discussed below, the CD domain does not make trimer-stabilizing contacts in our structure, further evidence of the important role played by the cap in the stabilization of the pre-fusion trimer.

## The 229E trimer can open at the bottom of the S2 subunit

A unique feature of our 229E S-protein trimer is the existence of a large opening that separates the monomers at the C-terminal end of the S2 subunit in the helical core and CD regions (*Figure 8a,b*).

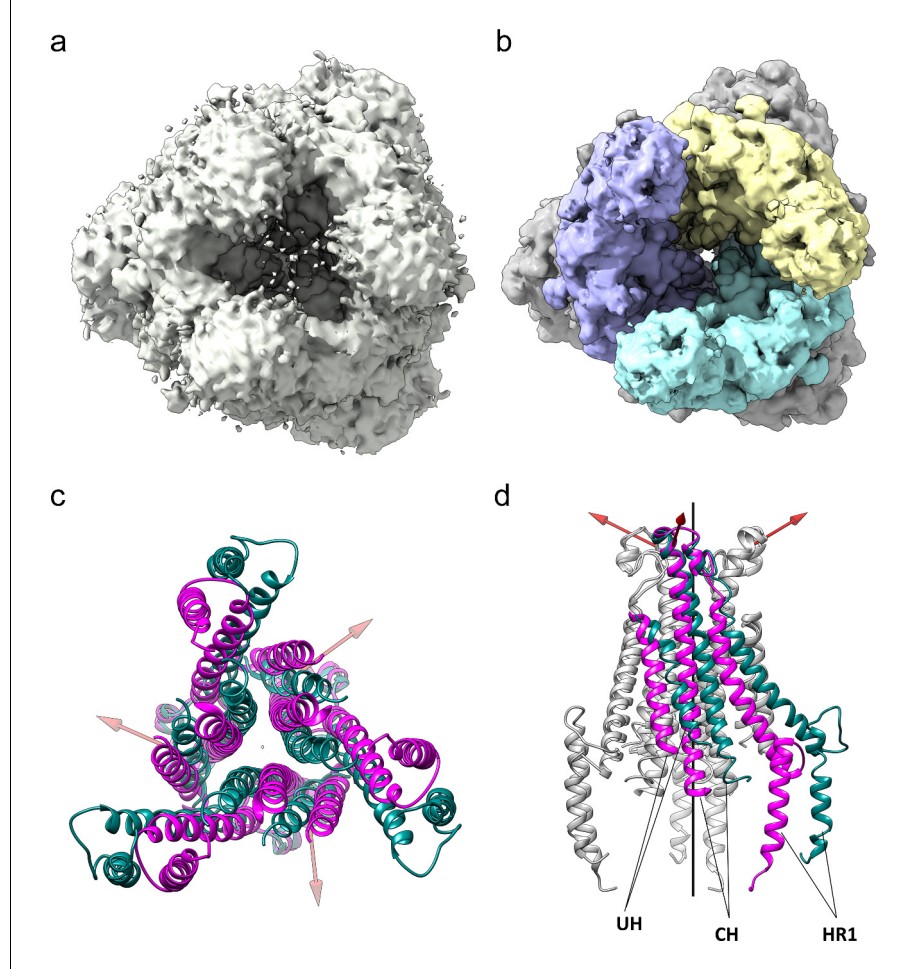

**Figure 8.** The HCoV-229E S-protein shows a large opening in the S2 region. (**a,b**) Bottom views of the S-protein trimer at two contour levels showing the opening in S2 at (**a**) 1.5 RMS and (**b**) 3.0 RMS. The three S2 subunits are colored differently in (**b**). (**c, d**) Comparison of the 229E (teal) and NL63 (magenta) S2 core helices. (**c**) Bottom view. (**d**) Side view, with only one subunit colored. Each 229E S2 subunit rotates ~11° relative to the NL63 S2 subunit. Red arrows indicate the rotation axes around which each monomer rotates.

The opening results in distinctive protrusions on the sides of the trimer that are clearly visible in the 2D class averages from both the negative stain and cryo-EM images (*Figure 2c,d*). Among other S-proteins only that of MHV (PDB:3JCL) shows a similar opening, albeit on a much smaller scale. For the closely related NL63 S-protein, the helical core and CD domains of each monomer are packed around the trimer axis for their entire length. A search of our cryo-EM images with a 229E homology model produced from the NL63 S-protein did not return particle images that lacked the opening.

Relative to NL63, the opening is described by the rotation of each monomer about an axis, one for each monomer, that is not co-linear with the trimer axis (*Figure 8c,d*). The three axes are positioned near the top of the S2 subunits and, as a result, the S2 subunits move little at the top end near the S1 cap, while moving considerably at the bottom end to create the opening. The most pronounced effect of the monomer rotations is a sliding of one monomer over another in the helical core region of the trimer interface (*Videos 1* and *2*). This leads to the exposure of a significant proportion of the CH, HR1$_{821-852}$ and UH surfaces to solvent, an outcome undoubtedly facilitated by the hydrophilic interfaces between them.

As we now show (*Figure 7c,d*, *Figure 7—figure supplement 1*), the hydrophilic nature of these interfaces is common among all the reported coronavirus S-protein structures, an observation suggesting a functional role. Conversion from the pre-fusion to the post-fusion form requires major

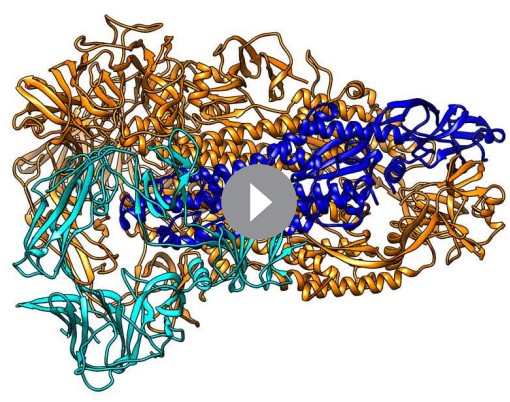

**Video 2.** Comparison of the 229E and NL63 S-proteins (Side view).
https://elifesciences.org/articles/51230#video2

fusion form.

conformational changes in the S2 region and our analysis suggests that these hydrophilic interfaces may play a role in the process. In the pre- and post-fusion structures of the MHV S-protein (PDB: 3JCL, 6B3O) (*Walls et al., 2016b*; *Walls et al., 2017*), the CH helix, the helix closest to the trimer axis, undergoes a 60° rotation along its long axis on conversion to the post-fusion form (*Video 3*). The rotation restructures the CH trimer interface and the transition is presumably facilitated by the hydrophilic interface as only a single apolar contact need be broken and formed in the process. Even larger conformational changes are associated with the formation of the post-fusion triple-helical HR1 coiled-coil (*Figure 5b,c*). It is certainly possible that the hydration of portions of the helical core region, shown to be possible by our structure, plays a role in this and other steps leading to the post-

## Discussion

Sequence analysis has shown that the three receptor binding loops of HCoV-229E are the most variable region in the entire viral genome. Indeed, loop variation has led to the emergence of six RBD classes whose viruses have successively replaced each other in the human population over the past 50 years (*Wong et al., 2017*). As we now show, the Class I, III, IV and V RBD-hAPN complexes show

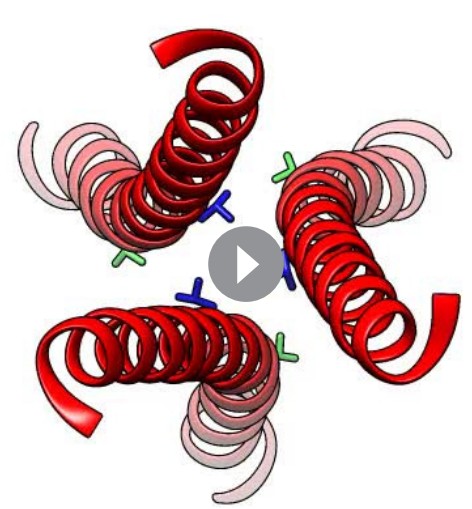

**Video 3.** Comparison of the CH helix in the pre-fusion and post-fusion conformations of the MHV S-protein. Top view showing the morphing between the two helices. PDB IDs: pre-fusion, 3JCL; post-fusion, 6B3O. Rotation of the CH helix during the conversion from the pre-fusion to the post-fusion conformation breaks one apolar contact (Leu 1062, blue) and forms another (Val 1070, green). The rest of the residues in the interfaces are polar.
https://elifesciences.org/articles/51230#video3

that there is a set of conserved core interactions, mediated by Loop 1 of the RBD, that defines the specificity of 229E for hAPN. The loop variation among classes is concentrated in regions peripheral to the core and loop plasticity ensures that they are accommodated while the core interactions are preserved. Immune evasion is a likely explanation for the emergence of these viral classes, and the structures provide insight into how mutational changes outside of the core region could abrogate loop-binding neutralizing antibodies while receptor binding is maintained.

The Class III, IV and V RBDs all bind hAPN with higher affinity than that of the tissue culture adapted Class I RBD (16-fold for Class V) (*Wong et al., 2017*). Given that the core interactions are conserved, the observed affinity changes undoubtedly arise from the sequence variation found in the periphery. Receptor binding affinity is a key determinant of viral fitness (*Hensley et al., 2009*; *de Vries et al., 2013*; *Lynch et al., 2015*; *Rockx et al., 2010*) and the optimization of it may be required following antibody escape or the acquisition of a new receptor during cross-species transmission. The structures also show how mutational changes in the periphery can lead to a modulation of receptor binding affinity while receptor specificity is maintained.

The nature of the interaction between the various RBDs and hAPN also provides insight into how HCoV-229E and the related bat, camel and alpaca 229E-like viruses were transmitted to their respective hosts. As the structures show, the common hydrogen bonds between the RBDs and hAPN all involve backbone atoms of hAPN. Moreover, the key 287–291 segment on hAPN is a surface exposed β-strand whose exposed backbone amide and carbonyl groups are satisfied by the hydrogen bonds with the RBD. The use of backbone atoms in this way reduces the dependence on a given sequence and increases the chances that a virus might productively bind the homologous receptor of another species in essentially the same way. The five Loop 1 residues that are absolutely conserved among all HCoV-229E viruses are also identical in the bat, camel and alpaca viruses (*Figure 1—figure supplement 1*). Taken together, it is likely that these viruses all use APN as receptor and that this has been facilitated by the Loop-1 mediated core interaction shown by our structures. During replication in the new host, loop mutations could accommodate the species-specific APN sequence differences in the peripheral regions and optimize receptor binding affinity as required. We have previously argued that the use of 'conserved' receptor interactions in this way would facilitate cross-species transmission over large evolutionary distances (*Wong et al., 2017*).

Shielding the receptor binding site from immune recognition is one explanation for why the RBDs of some coronavirus S-proteins are found to access more than one conformation (*Walls et al., 2016a*; *Gui et al., 2017*). In the down conformation, the receptor binding loops of 229E are blocked in the S-protein trimer where they cannot be bound by either B-cell receptors or antibodies. Whether they make S-protein contacts in the down conformation is an important question given the possibility that receptor binding loop variation might modulate the RBD equilibrium or the stability of the pre-fusion trimer. As shown by the HCoV-229E cryo-EM structure, the receptor binding loops of its Class I RBD closely approach domain A of the S1 subunit, but they do not interact with it strongly. The X-ray crystal structures of the Class III, IV and V RBDs in complex with hAPN provide structures for three other RBD classes, albeit in complex with hAPN. As shown by an overlay, they too do not interact significantly with domain A. In contrast, an analysis of the hinge region that connects domain B/RBD to domain C identifies two interfaces (*Figure 4h*) where interactions must be broken or formed in the conversion between the two RBD conformations. These interfaces are highly conserved among 229E isolates and taken together, it seems that the RBD equilibrium is a property of the pre-fusion trimer that is not likely to vary among viruses possessing different RBD classes.

It has also been suggested that receptor binding and conversion to the RBD up conformation might play a role in triggering the conversion to the post-fusion conformation (*Walls et al., 2019*; *Pallesen et al., 2017*). On conversion to the post-fusion conformation, the S2 subunit undergoes major conformational changes and the HR1-CH junction is restructured to form a continuous helix. These conformational changes are not possible in the presence of the S1 cap and domain B/RBD in the down conformation likely strengthens the S1 cap interaction with the S2 subunits through its contact with the HR1-CH junction. Our analysis suggests that the S1 cap is a stable assembly and to the extent that it is, the interaction between the cap and the S2 subunits would be subject to an avidity effect and weakest with all three RBDs in the up conformation. In this way, receptor engagement and an increase in the number of RBDs in the up conformation would facilitate conversion to the post-fusion form and viral entry.

As our cryo-EM structure shows, the HCoV-229E S-protein has an opening at the bottom end (*Figure 8*) that separates the monomers of the trimer in the helical core and CD regions of the S2 subunit. Relative to the NL63 trimer, a significant proportion of the 229E core helices, CH, $HR1_{821-852}$ and UH, are exposed to solvent. The interfaces between the monomers of the trimer in the helical core region are hydrophilic in nature and this has presumably facilitated the exposure of these helices to solvent. As we now show, this hydrophilic character is shared among the other coronavirus S-proteins for which structures exist (*Figure 7c,d*, *Figure 7—figure supplement 1*), an observation suggesting a conserved functional role. The coronavirus S2 subunit is large and the conformational changes on conversion to the post-fusion are more extensive than that shown by many other viral fusion proteins (*Harrison, 2015*; *Rey and Lok, 2018*). The HR1 triple helical coiled-coil, for example, is generated entirely on conversion to the post-fusion form. This is to be contrasted with the canonical HA of influenza virus where one-half of the triple helical coiled-coiled coil is already formed in the pre-fusion conformation. It is certainly possible that in the coronaviruses, these hydrophilic interfaces facilitate the S-protein conformational changes required in the conversion from the pre-fusion to the post-fusion forms.

## Materials and methods

### Cloning and expression of hAPN and the 229E RBDs

The RBD expression constructs and cell lines used were those described by *Wong et al. (2017)*. In that work, the Class I RBD was from sequence AAK32191.1. For the Class II - Class VI RBDs the receptor binding loop regions were from the following: ABB90507.1 (Class II), ABB90514.1 (Class III), ABB90519.1 (Class IV), ABB90523.1 (Class V), and AFR45554.1 (Class VI). The remainder of the RBDs comes from the Class I sequence. A sequence alignment of the six RBD classes can be found in *Figure 1—figure supplement 1*. The cDNA sequences encoding the six RBD classes were cloned into an inducible piggyBac transposon-based stable expression vector (*Li et al., 2013*). Stable cell lines were generated by co-transfecting HEK293S-GnT1(-/-) cells (ATCC Cat# CRL-3022, RRID: CVCL_A785) (*Reeves et al., 2002*) with each of the RBD plasmids and both a piggyBac transposase helper plasmid and an rtTA helper plasmid. The soluble ectodomain of hAPN (residues 66–967) was expressed in stably transfected HEK293S-GnT1(-/-) cells as described previously (*Wong et al., 2012*).

In all cases, the stable HEK293S-GnT1(-/-) cells were grown in Freestyle-293 medium (Gibco) in shaker flasks. The expression of the target protein was induced by the addition of 1 μg/mL doxycyclin. The target proteins were secreted as N-terminal protein-A fusion proteins with a TEV protease cleavage site following the protein-A tag. Harvested media were concentrated 10-fold and purified by IgG affinity chromatography (IgG Sepharose, GE). The bound proteins were liberated by on-column TEV protease cleavage. The hAPN was further purified by anion exchange chromatography (HiTrap-Q), while the HCoV-229E RBDs were further purified by cation exchange chromatography (HiTrap-SP). All of the cell lines used in this paper tested negative for mycoplasma contamination and were correctly identified as HEK293 cell lines.

### Crystallization of the 229E RBD-hAPN complexes

The purified hAPN was deglycosylated using the endoglycosidase Endo A (*Takegawa et al., 1997*). 0.5 mg of Endo A was added to as much as 15 mg of hAPN in 10 mL of buffer consisting of 10 mM MES pH 6.5 and 100 mM NaCl. This mixture was incubated at 37° C for 48 hr. After 48 hr, 20 μL of Jack Bean α-mannosidase (Sigma #M7257) was added and the pH was dropped to 5.0 by the addition of 30 μM sodium acetate. $ZnSO_4$ was also added to a final concentration of 1 mM. After an additional 48 hr, the deglycosylated hAPN was again purified using anion exchange chromatography (HiTrap-Q). The deglycosylated hAPN and the RBDs were then individually purified on a Superdex 200 column (GE Healthcare). To isolate the RBD-hAPN complexes, the RBDs and hAPN were mixed in a 1.2:1 molar ratio (RBD:hAPN) and again run on the Superdex 200 column. Before setting up the crystallization experiments, the complexes were concentrated to 10 mg/mL and Endo A was added to a final concentration of 0.1 mg/mL to deglycosylate the RBDs in situ. The protein solutions were mixed with well solution consisting of 9% PEG 8000, 1 mM oxidized glutathione, 1 mM reduced glutathione, 5% glycerol, and 100 mM MES, pH 6.5. The crystallization experiments were carried out by hanging drop vapor diffusion. Of the five classes tested for co-crystallization with hAPN, only the Class III, IV and V RBDs yielded crystals. Crystal quality was improved by macro-seeding. Crystals were cryo-protected with well solution containing 25% glycerol before vitrification.

### X-ray diffraction data collection and structure determination

The diffraction data for the Class V RBD-hAPN complex were collected on beamline 23ID-B at the Advanced Photon Source. Data for the Class III and IV RBD-hAPN complexes were collected on beamline 08ID-1 at the Canadian Light Source. Data reduction was performed using XDS (*Kabsch, 2010*) and HKL2000 (*Otwinowski and Minor, 1997*). The structures were solved by molecular replacement using a hAPN monomer (pdb id: 4FYQ) (*Wong et al., 2012*) and the Class I RBD after removal of the receptor binding loops (PDB ID: 6ATK (*Wong et al., 2017*), lacking residues 307–324, 348–359, and 403–410). The atomic models were built using Coot (*Emsley and Cowtan, 2004*) and iteratively refined using REFMAC5 (*Murshudov et al., 2011*) and Phenix.refine (*Afonine et al., 2012*). Data collection and refinement statistics can be found in *Supplementary file 2*.

## Cloning and expression of the 229E spike protein ectodomain

The Class I HCoV-229E sequence (AAK32191.1) was chosen for the cryo-EM analysis. The cDNA sequence encoding the ectodomain of the S-protein (residues 17–1113) was cloned into an inducible piggyBac transposon-based stable expression vector (*Li et al., 2013*). A T4 fibritin trimerization motif, a TEV cleavage site, and a 6xHis tag was added to the C-terminal end of the 229E ectodomain. In addition, Thr 871 and Ile 872 were both mutated to prolines to stabilize the pre-fusion form of the S-protein as previously described (*Pallesen et al., 2017*). A stable cell line was generated by co-transfecting Freestyle-293F cells (Thermo Fisher Scientific, RRID:CVCL_6642) with the resulting expression plasmids with both a piggyBac transposase helper plasmid and an rtTA helper plasmid. The stable cell line was then scaled-up in DMEM/F12 medium supplemented with 3%(v/v/) FBS in roller bottles. Protein expression was induced with the addition of 1 µg/mL doxycycline. The secreted 229E ectodomain was purified from the tissue culture medium using Ni-NTA resin (Qiagen). The protein was further purified by hydrophobic interaction chromatography using a RESOURCE ISO column (GE Healthcare). The protein was then dialyzed and further purified by size-exclusion chromatography using a Superdex 200 Increase column (GE healthcare) just prior to specimen preparation for electron microscopy.

## Negative-stain EM

3 µL of the S-protein ectodomain was applied to the surface of a glow-discharged (60 s, 15 mA), carbon-coated copper mesh grid. The grid was then washed with 150 mM NaCl and stained with 3 µL of 2% uranyl formate. Micrographs were acquired on either a Tecnai T20 200 kV electron microscope equipped with an Orius SC1000 CCD camera or on a Talos L120C 120 kV electron microscope equipped with a Ceta 16M CMOS camera. Particle selection and 2D classification were carried out in RELION2.2 (*Fernandez-Leiro and Scheres, 2017*) and cryoSPARC v2 (*Punjani et al., 2017*). Contrast transfer function (CTF) estimation was performed using GCTF (*Zhang, 2016*) or CTFFIND4 (*Rohou and Grigorieff, 2015*).

## Cryo-EM data collection

Holey gold film (*Russo and Passmore, 2014*) coated EM grids with 2 µm holes were prepared following a previously described procedure (*Marr et al., 2014*). The S-protein solution contained 0.4 to 0.8 mg/mL protein in 10 mM Tris, pH 8.0 and 150 mM NaCl. Vitrification of the cryo-EM specimen was performed using a Gatan Cryoplunge 3 device. For each cryo-EM specimen, 3 µL of S-protein solution was applied to a glow-discharged (2 min, 15 mA) holey grid. The grids were blotted for 13 s at 100% humidity and 295 K and then plunge-frozen in a 50:50 ethane:propane mixture (*Tivol et al., 2008*) at 77 K. Specimen screening and optimization was done with an FEI Tecnai TF20 electron microscope equipped with a Gatan K2 summit direct detector. The final high-resolution data collection was performed with a Thermo Fisher Scientific Titan Krios G3 300 kV microscope equipped with a Falcon 3EC direct detector. The final high-resolution data were collected at 75000x nominal magnification, resulting in a calibrated pixel size of 1.06 Å. Each movie was recorded in counting mode with a 60 s exposure and saved in 30 fractions. The exposure rate was adjusted to 0.8 electrons per pixel per second, resulting in a total exposure of 42.7 electrons per $Å^2$. The data were collected with a 1.8 to 2.2 µm defocus. A total of ~4000 movies were collected for the final high-resolution dataset.

## Cryo-EM data processing

Full-frame motion correction of the cryo-EM movies were initially performed using an implementation of the alignframes_lmbfgs algorithm (*Rubinstein and Brubaker, 2015*) in cryoSPARC v2 (*Punjani et al., 2017*). Local motion correction was later performed after initial particle processing in cryoSPARC v2 with an improved implementation of the alignparts_lmbfgs algorithm (*Rubinstein and Brubaker, 2015*). CTF parameters were estimated using CTFFIND4 (*Rohou and Grigorieff, 2015*). Particle selection was performed using multiple methods in both RELION2.2 (*Fernandez-Leiro and Scheres, 2017*) and cryoSPARC v2. The methods used included manual picking followed by template-guided automatic picking, and automatic picking with a Gaussian blob. For the template-guided auto-picking, templates were low-pass filtered to 20 Å. The micrographs all contained many small particles that did not correspond to the intact S-protein ectodomain. The particle images were therefore processed by multiple rounds of 2D-classification and 3D-classification to remove these

particles. During 2D-classification, only 2D classes with well-defined internal structures were selected for further processing. The final dataset included ~71,000 particle images. Initial map generation, homogenous 3D refinement, heterogenous 3D refinement and non-uniform 3D refinement were all carried out in cryoSPARC v2. The 3D refinements were carried out with either C1 or C3 symmetry imposed. Comparison of these results indicated that the 229E S-protein particles refined in C1 did not display significant non-three-fold symmetric features. Therefore, the maps generated from the C3 refinements were used for model building and analysis.

The maps showed well-resolved secondary structure and variable quality sidechain features for the entire S1 region and the majority of the S2 region. However, the C-terminal connector domain (starting at residue 959) showed weak density. The HR2 coiled-coil and the added foldon trimerization motif, which are further downstream of the connector domain, could be observed in the negative stain micrographs as long flexible tails, but they were not observed in the cryo-EM maps.

In an attempt to obtain better resolution for the poorly resolved connector domains, local refinement in the connector domain region was performed using cryoSPARC v2. For this purpose, soft masks were first generated for either one or three of the connector domains. Then the rest of the S-protein was subtracted from the particle images. Local refinement was then performed with either the mask of one connector domain or the mask including all three connector domains, following particle subtraction. Fulcrum positions were extensively tested and fine-tuned. Despite these efforts, the resolution improvement in the connector domain region was limited.

### S-protein ecto domain cryo-EM model building

The atomic model of the ectodomain of the HCoV-229E S-protein was built manually into the C3-symmetry map using Coot (*Emsley and Cowtan, 2004*). A homology model based on the 3.4 Å cryo-EM structure of the HCoV-NL63 S-protein trimer (PDB id: 5SZS) (*Walls et al., 2016a*) was produced using the PHYRE2 server (*Kelley et al., 2015*), and used as a reference during model building. Except for a few loops and the linker between S1 and S2, we were able to trace the polypeptide chain for residues 38–958. In the connector domain the homology model was docked into the map. This resulted in an atomic model of the HCoV-229E ectodomain that included most of the residues between residue 38 and residue 1033, 24 residues before the start of the HR2 region. The atomic model was refined against a half-map using Phenix.real_space_refine (*Afonine et al., 2018a*) and Rosetta (*Wang et al., 2016*). To improve the geometry of the atomic model, molecular dynamics-assisted model building was carried out in ChimeraX (*Goddard et al., 2018*) using the ISOLDE tool (*Croll, 2018*). The other half-map was used for generating model statistics. The atomic model was validated using Molprobity (*Williams et al., 2018*), EMRinger (*Barad et al., 2015*) and the comprehensive validation (cryo-EM) tool in Phenix (*Afonine et al., 2018b*). The molecular graphics, animations and movies were generated using UCSF Chimera (*Pettersen et al., 2004*) and UCSF ChimeraX (*Goddard et al., 2018*).

### Simulation of 229E cryo-EM particle images

The simulated 2D class average of the S1 cap shown in *Figure 2f* was generated using the 'Simulate Data' tool of cryoSPARC v2 (*Punjani et al., 2017*). A map portion corresponding to the S1 cap was isolated from the 229E cryo-EM map as input. 1000 simulated particle images were then generated using the 'Simulate Data' tool with the follow settings: defocus, 1.5 to 2.5 µm; acceleration voltage, 300 kV; amplitude contrast, 7%; spherical aberration, 2.7 mm; signal to noise ratio, 0.5; number of gridpoints for rotations, 3. The resulting simulated particle images were then used for 2D-classification. The resulting 2D class average corresponding to the top/bottom view of the S1 cap was used for comparison with the putative S1 cap images found in the cryo-EM 2D class averages shown in *Figure 2e*.

## Acknowledgements

The work was supported by CIHR operating grants to PJT, JLR and JMR, and a Canada Research Chair to JLR. The Canadian Light Source and the Advanced Photon Source at Argonne National Laboratory are acknowledged for synchrotron data collection. The Toronto High Resolution High Throughput cryo-EM facility, supported by the Canada Foundation for Innovation and Ontario Research Fund, is acknowledged for Cryo-EM data collection.

## Additional information

### Funding

| Funder | Author |
|---|---|
| Canadian Institutes of Health Research | James M Rini<br>John L Rubinstein<br>Pierre J Talbot |
| Canada Research Chairs | John L Rubinstein |

The funders had no role in study design, data collection and interpretation, or the decision to submit the work for publication.

### Author contributions

Zhijie Li, Conceptualization, Formal analysis, Validation, Investigation, Writing—original draft, Writing—review and editing; Aidan CA Tomlinson, Conceptualization, Investigation; Alan HM Wong, Marc Desforges, Conceptualization; Dongxia Zhou, Investigation; Pierre J Talbot, Conceptualization, Funding acquisition, Writing—review and editing; Samir Benlekbir, Supervision, Investigation; John L Rubinstein, Conceptualization, Supervision, Funding acquisition, Writing—review and editing; James M Rini, Conceptualization, Formal analysis, Supervision, Funding acquisition, Investigation, Writing—original draft, Writing—review and editing

### Author ORCIDs

Zhijie Li (iD) http://orcid.org/0000-0001-9283-6072
John L Rubinstein (iD) http://orcid.org/0000-0003-0566-2209
James M Rini (iD) https://orcid.org/0000-0002-0952-2409

### Decision letter and Author response

Decision letter https://doi.org/10.7554/eLife.51230.sa1

## Additional files

### Supplementary files

• Supplementary file 1. Cryo-EM data collection and refinement statistics.

• Supplementary file 2. X-ray crystallographic data collection and refinement statistics. **Values in parentheses are for the highest-resolution shell.

• Transparent reporting form

### Data availability

The X-ray diffraction data and X-ray crystal structures have been deposited in PDB under accession codes 6U7E, 6U7F and 6U7G. The cryo-EM map has been deposited in EMDB under accession code EMD-20668. The cryo-EM structure has been deposited in PDB under accession code 6U7H.

The following datasets were generated:

| Author(s) | Year | Dataset title | Dataset URL | Database and Identifier |
|---|---|---|---|---|
| Li Z, Tomlinson ACA, Wong AHM, Zhou D, Desforges M, Talbot, PJ, Benlekbir S, Rubinstein JL, Rini JM | 2019 | X-ray diffraction data and X-ray crystal structures | https://www.rcsb.org/structure/6U7E | RCSB Protein Data Bank, 6U7E |
| Li Z, Tomlinson ACA, Wong AHM, Zhou D, Desforges M, Talbot, PJ, Benlekbir S, Rubinstein JL, Rini JM | 2019 | X-ray diffraction data and X-ray crystal structures | https://www.rcsb.org/structure/6U7F | RCSB Protein Data Bank, 6U7F |

| Li Z, Tomlinson ACA, Wong AHM, Zhou D, Desforges M, Talbot, PJ, Benlekbir S, Rubinstein JL, Rini JM | 2019 | X-ray diffraction data and X-ray crystal structures | https://www.rcsb.org/structure/6U7G | RCSB Protein Data Bank, 6U7G |
| --- | --- | --- | --- | --- |
| Li Z, Tomlinson ACA, Wong AHM, Zhou D, Desforges M, Talbot, PJ, Benlekbir S, Rubinstein JL, Rini JM | 2019 | cryo-EM structure | https://www.rcsb.org/structure/6U7H | RCSB Protein Data Bank, 6U7H |
| Li Z, Tomlinson ACA, Wong AHM, Zhou D, Desforges M, Talbot, PJ, Benlekbir S, Rubinstein JL, Rini JM | 2019 | cryo-EM map | https://www.ebi.ac.uk/pdbe/entry/emdb/EMD-20668 | Electron Microsocy Data Bank, EMD-20668 |

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
