## [Decision Letter]

[Editors’ note: this was accepted after the first round of peer review, so there is not an accompanying Author response.]

This manuscript from Li et al reports the HCoV‐229E S protein ectodomain structure and several structures of the receptor-binding domains from different RBD classes bound to the receptor hAPN. The latter structures provide a molecular basis for how sequence variation in the RBDs can be accommodated while maintaining a conserved interaction with hAPN. The cryo-EM structure of the HCoV‐229E S protein ectodomain in the prefusion conformation provides new insights into the arrangement and interactions of the S2 subunit, with implications for the conformational changes that are undergone during refolding of S2 to the post-fusion conformation.

Strengths of the manuscript are the clarity of the writing, quality of the structures, and presentation of the results, including movies. The three crystal structures and the cryo-EM structure are excellent, as are the cryo-EM data processing and interpretation. It is appreciated that the validation reports and EM maps were provided for review. The results and conclusions represent solid contributions to the coronavirus field and provide the first evidence for an arrangement of the S2 subunits that is substantially more 'open' than those observed in other S protein structures. Only a few minor suggestions are noted below.

Minor Comments:

1) The first paragraph of the Results section is redundant with the second-to-last paragraph of the Introduction. I would suggest removing it and replacing it with a single introductory sentence.

2) The authors introduce a domain nomenclature (D1, D2, D3...) that is different than the two currently used in the field. Using one of the previous systems may lead to less confusion for readers.

3) The Discussion describes the authors' analysis and interpretation of the RBD conformational changes and how 3 RBDs in the up conformation would be unstable and lead to conversion to the post-fusion conformation. It should be noted that this same analysis and interpretation were presented in Pallesen et al., 2017, shown in Figure 7.